# Meta-analyses of *Schistosoma japonicum* infections in wild rodents across China over time indicates a potential challenge to the 2030 elimination targets

**Hui-Ying Zou**[1,2]**, Qiu-Fu Yu**[1,2]**, Chen Qiu**[1,2]**, Joanne P. Webster**[2,3]**, Da-Bing Lu**[1,2]*

**1** Department of Epidemiology and Statistics, School of Public Health, Soochow University, Suzhou, China, **2** Key Laboratory of National Health and Family Planning Commission on Parasitic Disease Control and Prevention, Jiangsu Provincial Key Laboratory on Parasite and Vector Control Technology, Jiangsu Institute of Parasitic Diseases, Wuxi, China, **3** Centre for Emerging, Endemic and Exotic Diseases (CEEED), Department of Pathology and Population Sciences, Royal Veterinary College, University of London, London, United Kingdom

* Ludabing@suda.edu.cn

**Data Availability Statement:** All relevant data are within the manuscript and its Supporting Information files.

## Abstract

China once suffered greatly from schistosomiasis japonica, a major zoonotic disease. Nearly 70 years of multidisciplinary efforts have achieved great progress in disease control, with infections in both humans and bovines significantly reduced to very low levels. However, reaching for the target of complete interruption of transmission at the country level by 2030 still faces great challenges, with areas of ongoing endemicity and/or re-emergence within previously 'eliminated' regions. The objectives of this study were, by using meta-analytical methods, to estimate the overall prevalence of *Schistosoma japonicum* infections in abundant commensal rodent species in mainland China after the introduction of praziquantel for schistosomiasis treatment in humans and bovines in 1980s. In doing so we thereby aimed to further assess the role of wild rodents as potential reservoirs in ongoing schistosome transmission. Published studies on infection prevalence of *S. japonicum* in wild rodents in mainland China since 1980 were searched across five electronic bibliographic databases and lists of article references. Eligible studies were selected based on inclusion and exclusion criteria. Risks of within and across study biases, and the variations in prevalence estimates attributable to heterogeneities were assessed. The pooled infection prevalence and its 95% confidence intervals (CIs) were calculated with the Freeman-Tukey double arcsine transformation. We identified a total of 37 relevant articles involving 61 field studies which contained eligible data on 8,795 wild rodents across mainland China. The overall pooled infection prevalence was 3.86% (95% CI: 2.16–5.93%). No significant change in the overall pooled prevalence was observed between 1980–2003 (n = 23 studies) and 2004-current (n = 38 studies). However, whilst the estimated prevalence decreased over time in the marshland and lake regions, there was an apparent increase in prevalence within hilly and mountainous regions. Among seven provinces, a significant prevalence reduction was only seen in Jiangsu where most endemic settings are classified as the marshland and lakes. These estimates changed over season, ranging from 0.58% in spring

**Funding:** The authors were funded by the National Science Foundation of China (to DL, 81971957), by the Jiangsu Provincial Project of Invigorating Health Care through Science, Technology and Education (to DL, wk018-001), and by a ZELS research grant (combined BBSRC, MRC, ESRC, NERC, DSTL & DFID: BB/L018985/1 to JPW). The funders had no role in study design, data collection and analysis, decision to publish, or preparation of the manuscript.

**Competing interests:** The authors have declared that no competing interests exist.

to 22.39% in winter, in association with increases in rodent density. This study systematically analyzed *S. japonicum* infections in wild rodents from the published literature over the last forty years after the introduction of praziquantel for schistosomiasis treatment in humans and bovines in 1980s. Although numbers of schistosomiasis cases in humans and bovines have been greatly reduced, no such comparable overall change of infection prevalence in rodents was detected. Furthermore, there appeared to be an increase in *S. japonicum* prevalence in rodents over time within hilly and mountainous regions. Rodents have been projected to become the dominant wildlife in human-driven environments and the main reservoir of zoonotic diseases in general within tropical zones. Our findings thus suggest that it is now necessary to include monitoring and evaluation of potential schistosome infection within rodents, particularly in hilly and mountainous regions, if we are ever to reach the new 2030 elimination goals and to maximize the impact of future public, and indeed One Health, interventions across, regional, national and international scales.

## Author summary

Consistent with the revised WHO Global Goals, China has set the target of complete interruption of transmission (elimination) of zoonotic *Schistosoma japonicum* by 2030 at the entire country level. Much remains to be known, however, regarding the complex multi-host disease dynamics of schistosomiasis. Multi-disciplinary disease control programs within China, including mass treatment with praziquantel for over 40 years, have successfully targeted human and bovine definitive hosts, significantly reducing infection prevalence to extremely low levels. However, *S. japonicum* has at least 40 species of potential definitive host reservoirs. Most notably, and perhaps most challenging in terms of targeted control, high prevalence of *S. japonicum* has been detected in rodent wildlife within recent years, particularly in hilly and mountainous regions of China. Molecular/phylogenetic studies have revealed matched *S. japonicum* genotypes indicative of shared transmission between rodents and humans. Similarly, mathematical models, incorporating parasitological and molecular data, have indicated that rodents may be sufficient to maintain ongoing transmission of schistosomiasis within some Chinese regions, most notably that of hilly and mountainous habitats. In order to help elucidate further the potential association between prevalence of *S. japonicum* in humans with rodent wildlife across China, and to assess whether this balance may be changing following the introduction of mass drug administration with praziquantel to humans (and bovines) in the 1980s, we performed a meta-analyses aimed to estimate the overall prevalence of *S. japonicum* infections in commensal species rodents in China over this period. Published studies on *S. japonicum* infections in wild rodents in China since 1980 were searched for across five electronic bibliographic databases, together with lists of cited articles. We identified a total of 37 relevant articles involving 61 studies with eligible data on 8795 rodents. The pooled prevalence level was 3.86% (95% CI: 2.16–5.93%). No significant change in overall pooled prevalence levels within rodents was observed between 1980–2003 and 2004-current, despite the integrated control strategies performed across China within the latter period. However, whilst the prevalence estimates did decrease within marshland and lake regions since 2004, in the hilly and mountainous regions, in contrast, there was a significant increase in the rodent infection prevalence. The estimate increased from the smallest 0.58% in spring to the highest 22.39% in winter. It also increased with the density of

rodents. Therefore, we suggest that future systematic monitoring of schistosome infection levels within potential wildlife reservoirs, particularly within hilly and mountainous regions within China and/or areas aiming for verification of interruption of transmission, should be incorporated in order to reliably evaluate impact, risk and ultimately help ensure the sustainability of elimination interventions across, regional, national and international scales.

## Introduction

Schistosomiasis is the second most important parasitic disease after malaria, in terms of socio-economic impact, and is endemic in 78 tropical and subtropical countries. It is estimated that over 220 million people are currently infected with schistosomes, with more than 70 million new infections and thousands of deaths occurring annually [1]. The majority of human infections and morbidity are caused by three main schistosome species: *Schistosoma mansoni*, *S. haematobium*, and *S. japonicum* [2], among which *S. japonicum* is the only human blood fluke that is endemic in China [3]. China once profoundly suffered from schistosomiasis, termed 'the God of Plague' [4]. At the beginning of the national schistosomiasis control programme in the 1950's, approximately 100 million Chinese people (of a total population of approximately 600 million) were at risk of schistosome infection, and an estimated 11.612 million people were infected [5]. Nearly 70 years of integrated multi-disciplinary control efforts, including mass drug administration (MDA) of praziquantel (PZQ) to both humans and bovine hosts since the 1980s [6], has achieved tremendous progress in reducing prevalence and intensity levels of human and bovine schistosomiasis [7]. Recent (2017) surveillance data reported prevalences down to 0.002% in humans and 0 in bovines [8]. As a consequence, China, consistent with the revised WHO Global Goals, has set the target of complete interruption of transmission (elimination) of zoonotic *S. japonicum* by 2030 at the entire country level [9].

However, schistosomiasis japonica still remains a public health concern in China [3, 10]. By the end of 2017, a total of 82 out of 450 counties (cities or districts) had not achieved the level of interruption [8], and some previously 'eliminated' areas have observed recrudescence of the disease [11]. Many of these persistent and/or re-emerging regions are distributed across seven provinces including Hunan, Hubei, Jiangxi, Anhui and Jiangsu, which are located in the middle and lower reaches of the Yangtze River, as well as the two mountainous provinces of Yunnan and Sichuan. In 2017 a total area of 622 454.49 hm$^2$ was surveyed and *S. japonicum* intermediate host snails, *Oncomelania hupensis*, were found across an area of 172 501.56 hm$^2$, out of which newly detected snail areas covered up to 208.54 hm$^2$ [8] (as well as suitable habitats for the alarming recent rapid expansion of *Biomphalaria straminea*, the intermediate host for *S. mansoni* in the southern China [12]). Under such conditions of extensive snail habitats, any re-seeding of *S. japonicum* from either imported or sympatric definitive hosts, of any species, are likely to results in new and/or maintained transmission. Indeed, one of the greatest challenges for interruption of transmission is that schistosomiasis japonica is a multi-host zoonotic disease, with at least 40 known species of mammalian definitive hosts [13]. Of these, both humans and bovines were traditionally accepted to be primarily responsible for the ongoing transmission of schistosomiasis across China [14, 15]. However, there is a growing body of evidence suggesting there are currently high levels of *S. japonicum* infections in small rodents in some areas, particularly hilly/mountainous regions [16–18]. Rodents have a wide distribution and high reproduction potential, with efforts for their control proving notoriously challenging worldwide. The commensal and sympatric nature of rodents places them in close human (and

domestic livestock) contact, and the widespread distribution of rodent faeces around ditches and snail habitats provide ideal locations for miracidal hatching and subsequent ongoing transmission. Of key importance is molecular phylogenetic work (including following hatching of viable miracidia) demonstrating it is the same *S. japonicum* shared genotypes circulating through rodents, humans and bovines in China [19–22] (as has also recently been demonstrated between humans and rodents in Africa [23]). Finally, mathematical models, based upon epidemiological, parasitological and molecular data, have revealed that whilst $R_o$ levels within humans are now sufficiently low within China that interruption of transmission could be achieved if schistosomiasis japonicum were an exclusively human disease, rodents appear to be maintaining transmission in certain hilly/mountainous regions, whilst bovines appear responsible in lowland/marshy regions [17].

Therefore, in this study we performed a meta-analysis of the infection prevalence of *S. japonicum* in rodents in China reported after the introduction of praziquantel for treatment in 1980s and following additional major changes in integrated national control programme activities. Our objective was to elucidate whether estimates in the prevalence of *S. japonicum* in wild rodents mirrored the ongoing downward trend of *S. japonicum* infection prevalence in humans and livestock. By doing so we aimed to determine any change in the role of rodents as reservoir or spill-over hosts, or even possible key hosts (see e.g. [24] for definitions), and thus help evaluate the current potential for China reaching complete interruption of schistosomiasis transmission by 2030 [9, 25].

## Methods

### Search strategy and selection criteria

We searched five electronic bibliographic databases for relevant publications after 1980. 1980 was chosen as baseline as this was the date that praziquantel distribution to humans and animals commenced within China [6]. Furthermore, during the Cultural Revolution of 1966–1976, very few scientific papers, if any, were available on this topic. Likewise, of the few papers published during 1950's and/or between 1976–1980, insufficient information was provided on prevalence levels and thus were excluded here. In 2004, the central government of China classified schistosomiasis as one of the highest priorities in infectious diseases control [26] and a revised medium- and long-term control plan was then developed [27]. For this reason, we further sub-divided part of our analyses into 1980–2003 and then 2004-current. Chinese databases such as CNKI, VIP and Wan fang were retrieved by using "xuexichong and/or xuexichongbing and yeshu" as Chinese keywords. "*Schistosoma japonicum* and/or schistosomiasis japonica and rodents and china" were used as the key words to search PubMed and Web of Science. We also searched relevant reference lists and relevant journals by hand. Our analyses were accorded with the preferred reporting items for systematic reviews and meta-analyses (PRISMA) guidelines [28] for a systematic review of prevalence (S1 PRISMA Checklist).

Literature selection criteria: (1) provided full texts; (2) were not republished or with duplicated data; (3) performed in endemic areas within China; (4) provided geographical location, at least specific to provinces; (5) belonged to field investigations; (6) reported the time performed; (7) reported numbers of investigated and infected rodents, or could be calculated by formula; (8) with sample size of more than ten. Studies were excluded if they did not fulfill any of these criteria.

Two reviewers HZ and QY collected the data separately, including selection criteria, data extraction and statistical methods. Where data were inconsistent, a third examiner CQ would look for the cause and resolve the problem.

### Data extraction

From each eligible study, the following data were extracted: the first author, year of publication, year of study, location, seasons, rodent density (the number captured/the number of the rodent traps) and species, eco-epidemiological endemic setting, and numbers of the infected and dissected rodents.

### Data analysis

All extracted data were entered and transferred into the Meta package in R3.5.2 for statistical analysis. The pooled infection prevalence and its 95% confidence intervals (CIs) of *S. japonicum* in rodents were calculated with the Freeman-Tukey double arcsine transformation [29, 30]. Pooled prevalence (and CIs) is obtained via aggregation of the results of multiple studies. This aggregation is therefore not just the simple sum of the data obtained from all the studies, but a procedure that 'weights' the results of each study according to its precision. The precision of each study is estimated on the width of the dispersion (i.e. variance), and then the weight of each study is given by the inverse of the variance. Thus, if a study has a wide variance it will have a small weight in determining the final result of the analysis (the pooled prevalence of the meta-analysis), while a study with a small variance will have a greater weight [31]. The double arcsine transformation used here addresses both the problem of confidence limits when calculated possibly outside the 0..1 range and that of variance instability caused by any prevalence of close to 0 or 1 [29]. Heterogeneity was quantified using the inverse variance statistic ($I^2$ index) and tested for significance by the Cochran Q test, in which case p-values were statistically significant at $p \leq 0.05$. An $I^2$ index was interpreted as low, moderate or high heterogeneity if it had a value of $\leq 25\%$, $\leq 50\%$, or $> 75\%$, respectively [32, 33]. When there was evidence of heterogeneity ($I^2 > 50\%$), infection prevalence were pooled by using a random-effects model; otherwise, prevalence was pooled by using a fixed-effects model [34].

A forest plot was generated to visualize prevalence data among included studies. We performed subgroup analyses based on study period (i.e. before and after 2004, since the year of 2004 an integrated control strategy has been performed [35]), location, season, endemic setting, or rodent density (and rodent species, if possible). To evaluate any potential publication bias, a funnel graph was generated for each estimate, and then was statistically evaluated with the Egger test [36] with Stata SE15.1. A two-tailed p value < 0.05 was considered statistically significant.

## Results

### Literature search

Search results are shown in Fig 1. We retrieved a total of 1,238 records. After removing duplicates and preliminary screening, we comprehensively reviewed 95 articles. After excluding a further 58 unqualified reports including 12 reviews, 7 repeated data, 4 no study time, 3 no field survey, 5 no infection data, 1 no study location, 22 not in endemic regions and 4 sample size of less than ten, a total of 37 articles (34 in Chinese and 3 in English) [16, 37–72] were included. As one article may report one or more studies, this meta-analysis then included 61 studies covering seven provinces.

### Study characteristics

Among the included 61 studies, 23 studies were conducted during 1980 to 2003 and 38 during 2004 to 2018. A total of 17 studies were performed in Anhui province, 13 in Hubei, 8 in Jiangsu, 11 in Yunnan, 7 in Hunan, 3 in Sichuan and 2 in Jiangxi. A total of 8795 wild rodents

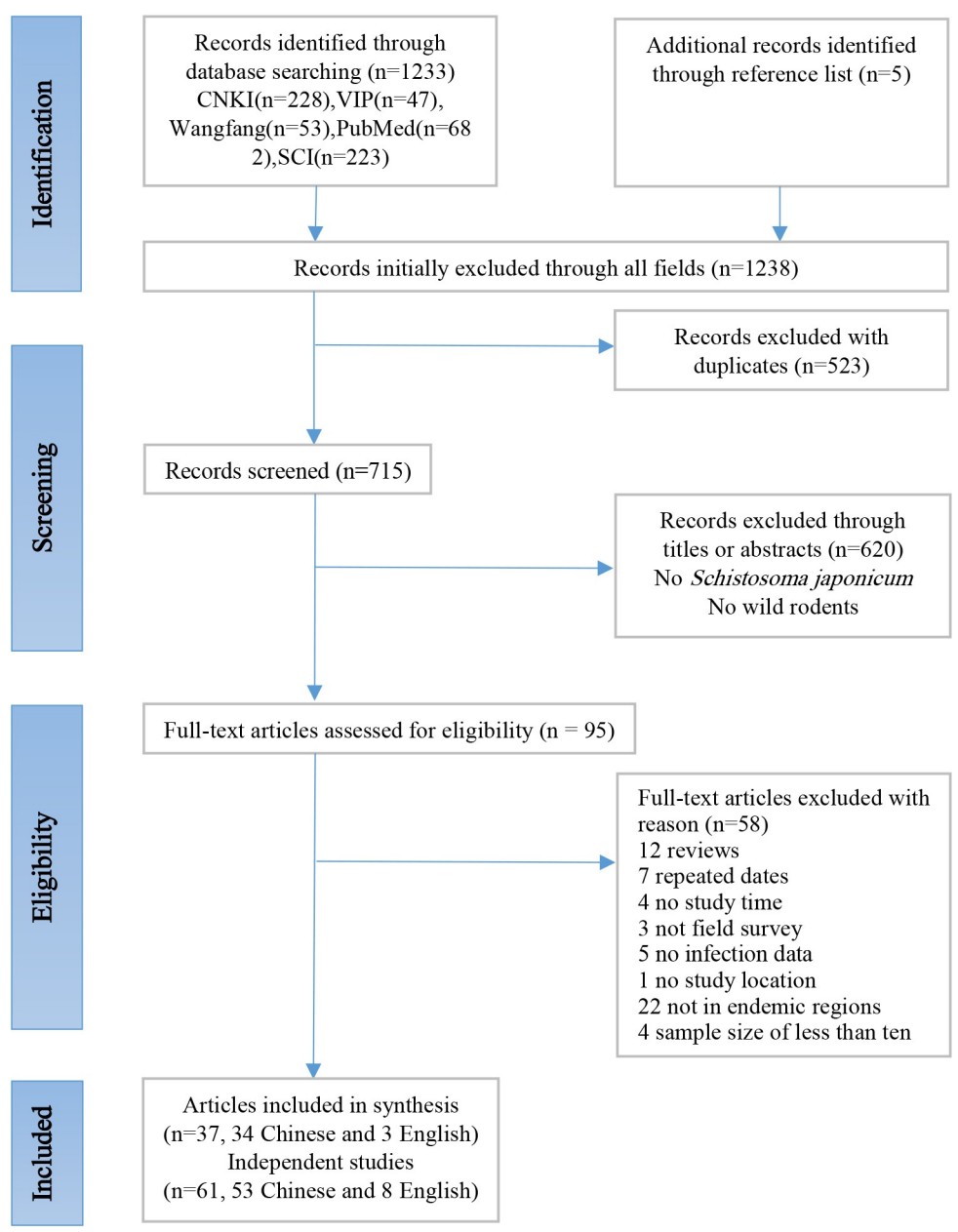

**Fig 1. Flow chart of study search and selection strategy.** The flow diagram shows the numbers of titles and studies reviewed in preparation of this meta-analysis of *S. japonicum* infection prevalence in rodents across China.

were investigated and 260 were identified with *S. japonicum* infection. The infection prevalence in rodents among studies ranged from 0 to 64.15% (Table 1).

## Pooling and heterogeneity analysis

Fig 2 shows the forest plot of infection prevalence levels in wild rodents. There was high heterogeneity among all studies and among studies within most subgroups (Table 2). By using a random-effects model, the overall pooled infection prevalence was 3.86% (95% CI: 2.16–5.93%). Fig 3 and Table 2 show the estimates grouped by potential influential factors. No

**Table 1. Characteristics of the eligible studies.**

| Author and year of publication* | Year of study performed | Province | Eco-epidemiological settings | Name of the pilot area | Season | Density of rodents | Detection technique | No. of examined | No. of infected (%) |
|---|---|---|---|---|---|---|---|---|---|
| Lu 2019[37] | 2018 | Jiangxi | HM | - | - | - | Dissection | 172 | 5(2.91) |
| Xu 2019[38] | 2015–2018 | Sichuan | HM | - | Autumn | - | Dissection | 59 | 0(0.00) |
| Zhang 2019[39] | 2018 | Yunnan | HM | Qiandian | - | - | Dissection and FE | 320 | 0(0.00) |
| | 2018 | Yunnan | HM | Wenbi | - | - | Dissection and FE | 126 | 0(0.00) |
| | 2011 | Yunnan | HM | Qiandian | - | - | Dissection and FE | 210 | 2(0.95) |
| | 2011 | Yunnan | HM | Wenbi | - | - | Dissection and FE | 165 | 0(0.00) |
| Kong 2018[40] | 2013 | Hubei | LM | - | Spring | 7.72% | Unclear | 67 | 0(0.00) |
| | 2013 | Hubei | LM | - | Autumn | 5.22% | Unclear | 46 | 0(0.00) |
| Lu 2018 [41] | 2017 | Jiangxi | HM | Village 3 | Summer | 6.56% | Dissection and MHT | 16 | 0(0.00) |
| Wang 2018[42] | 2017 | Hubei | LM | - | - | 3.84% | Dissection | 66 | 0(0.00) |
| Li 2018[43] | 2015–2016 | Hubei | LM | - | Autumn | - | Dissection | 49 | 2(4.08) |
| Van Dorssen 2017 [44] | 2014 | Hunan | LM | - | - | - | Dissection and FE | 83 | 0(0.00) |
| Shao 2016[45] | 2010–2011 | Yunnan | HM | - | - | 10.87% | Dissection and FE | 261 | 1(0.38) |
| Zuo 2016[46] | 2016 | Jiangsu | LM | - | Winter | 1.94% | Dissection | 62 | 0(0.00) |
| Zhang 2014[47] | 2013 | Hubei | LM | - | Spring | 9.08% | Dissection | 67 | 2(2.99) |
| Luo 2014[48] | 2012 | Hubei | LM | - | Spring | 10.59% | Dissection | 34 | 0(0.0) |
| Liu 2013[16] | 2011 | Anhui | HM | Sankouzhen | Autumn | 9.86% | Dissection and FE | 14 | 0(0.0) |
| | 2011 | Anhui | HM | Ducun | Autumn | 20.42% | Dissection and FE | 49 | 6(12.24) |
| Guo 2013[49] | 2011 | Hunan | LM | | Summer | - | Dissection | 51 | 7(13.72) |
| | 2011 | Hunan | LM | - | Summer | - | Dissection | 51 | 2(3.92) |
| | 2011 | Hunan | LM | | Summer | - | Dissection | 39 | 5(12.82) |
| | 2011 | Hunan | LM | | Summer | - | Dissection | 19 | 2(10.53) |
| Shao 2011[50] | 2010 | Yunnan | HM | - | - | 13.55% | Dissection and FE | 157 | 0(0.00) |
| He 2011[51] | 2010 | Hubei | LM | - | - | 7.11% | Dissection | 124 | 0(0.00) |
| Xia 2011[52] | 2008–2009 | Anhui | HM | - | - | - | Dissection and FE | 25 | 4(16.00) |
| | 2008–2009 | Anhui | HM | - | - | - | Dissection and FE | 23 | 3(13.04) |
| | 2008–2009 | Anhui | HM | - | - | - | Dissection and FE | 54 | 0(0.00) |
| Zhang 2010[53] | 2004–2009 | Jiangsu | LM | - | - | 1.28% | Dissection | 61 | 1(1.64) |
| Lu 2010[54] | 2007 | Anhui | HM | Longquan | - | - | Dissection and FE | 12 | 4(33.33) |
| | 2007 | Anhui | HM | Longshang | - | - | Dissection and FE | 22 | 3(13.63) |
| | 2007 | Anhui | HM | Yuantou | - | - | Dissection and FE | 17 | 2(11.76) |
| Ding 2008[55] | 2007 | Hubei | LM | - | - | - | Dissection | 62 | 12(19.35) |
| | 2007 | Hubei | LM | Taohua | - | - | Dissection | 59 | 7(11.86) |
| | 2007 | Hubei | LM | - | - | - | Dissection | 52 | 6(11.54) |

(*Continued*)

**Table 1.** (Continued)

| Author and year of publication* | Year of study performed | Province | Eco-epidemiological settings | Name of the pilot area | Season | Density of rodents | Detection technique | No. of examined | No. of infected (%) |
|---|---|---|---|---|---|---|---|---|---|
| | 2007 | Hubei | LM | - | - | - | Dissection | 43 | 5(11.63) |
| Wang 2007[56] | 2006 | Anhui | HM | - | - | - | Dissection and FE | 43 | 12(27.91) |
| Lu 2007[57] | 2006 | Anhui | HM | Longquan | Winter | 12.76% | Dissection and FE | 18 | 6(33.33) |
| | 2006 | Anhui | HM | Longshang | Winter | 15.49% | Dissection and FE | 22 | 5(22.73) |
| Gu 2001[58] | 1996–1998 | Sichuan | HM | - | - | - | Dissection | 72 | 0(0.00) |
| Yang 2000 [59] | 1995 | Jiangsu | LM | Xiaba | - | - | Dissection | 23 | 0(0.00) |
| YangX 1999[60] | 1997 | Yunnan | HM | - | Summer | - | Dissection and FE | 973 | 3(0.31) |
| YangW 1999[61] | 1997–1998 | Yunnan | HM | - | Summer | 10.17% | Dissection and FE | 1866 | 3(0.18) |
| Xu 1999[62] | 1996–1997 | Jiangsu | LM | - | Winter | - | Dissection | 69 | 43(62.32) |
| | 1997–1998 | Jiangsu | LM | - | Winter | - | Dissection | 53 | 34(64.15) |
| | 1997–1998 | Jiangsu | LM | - | Winter | - | Dissection | 67 | 36(53.73) |
| Wang 1997[63] | 1992–1995 | Anhui | LM | - | - | - | Dissection | 120 | 10(8.33) |
| Lu 1997[64] | 1993 | Anhui | LM | - | - | - | Dissection | 28 | 3(10.71) |
| | 1994 | Anhui | LM | - | - | - | Dissection | 11 | 1(9.09) |
| | 1995 | Anhui | LM | - | - | - | Dissection | 31 | 0(0.00) |
| Li 1996[65] | 1993 | Hubei | LM | - | - | - | Dissection | 101 | 0(0.00) |
| Zhou 1996[66] | 1993 | Hunan | LM | - | - | - | Unclear | 230 | 0(0.00) |
| | 1992 | Hunan | LM | - | - | 14.26% | Unclear | 405 | 16(3.95) |
| Yang 1995[67] | 1992 | Jiangsu | LM | Xinmin | Winter | - | Dissection | 62 | 0(0.00) |
| | 1993 | Jiangsu | LM | Xinmin | Winter | - | Dissection | 30 | 0(0.00) |
| Xu 1995[68] | 1987–1989 | Sichuan | HM | - | - | 12.30% | Dissection | 115 | 1(0.87) |
| Wang 1995[69] | 1993 | Anhui | LM | Guanghui | - | - | Dissection | 50 | 6(12.00) |
| Qiu 1995[70] | 1990–1992 | Yunnan | HM | Ziyou | - | - | Dissection | 431 | 0(0.00) |
| | 1990–1992 | Yunnan | HM | Banju | - | - | Dissection | 590 | 0(0.00) |
| | 1990–1992 | Yunnan | HM | Hedong | - | - | Dissection | 43 | 0(0.00) |
| Su 1994[71] | 1989–1990 | Hubei | LM | - | - | - | Dissection and FE | 21 | 0(0.00) |
| Shen 1986[72] | 1981–1983 | Anhui | HM | - | - | - | Dissection | 584 | 0(0.00) |

Note

*The included articles were listed in the order of publication year.

Abbreviations: HM, Hilly and mountainous regions; LM, Lake and marshland regions. FE, the fecal examination Kato-katz for eggs. MHT, the miracidial hatching technique.

obvious change was found between two periods, 1980 to 2003 (3.90%, 95% CI: 1.27–7.58%) and 2004 to 2018 (3.73%, 95% CI: 1.81–6.14%). Further divided by eco-epidemiology settings, as seen in Table 3, the estimate since 2004 had non-significantly decreased in marshlands and lakes but significantly (p<0.0001) increased in the hilly and mountainous regions. At the level of provinces, a significant (p = 0.04) and rapid reduction in the estimates, from 21.43% during 1980 to 2003 to 0.51% during 2004 to 2018, was seen in Jiangsu. Other provinces saw a significant (in Hubei, p = 0.03) or non-significant increase (in Hunan, Anhui and Yunnan with p = 0.17, 0.10 and 0.39, respectively). In terms of season or rodent density, the pooled infection prevalence of *S. japonicum* was low in spring (0.58%), increased throughout summer and

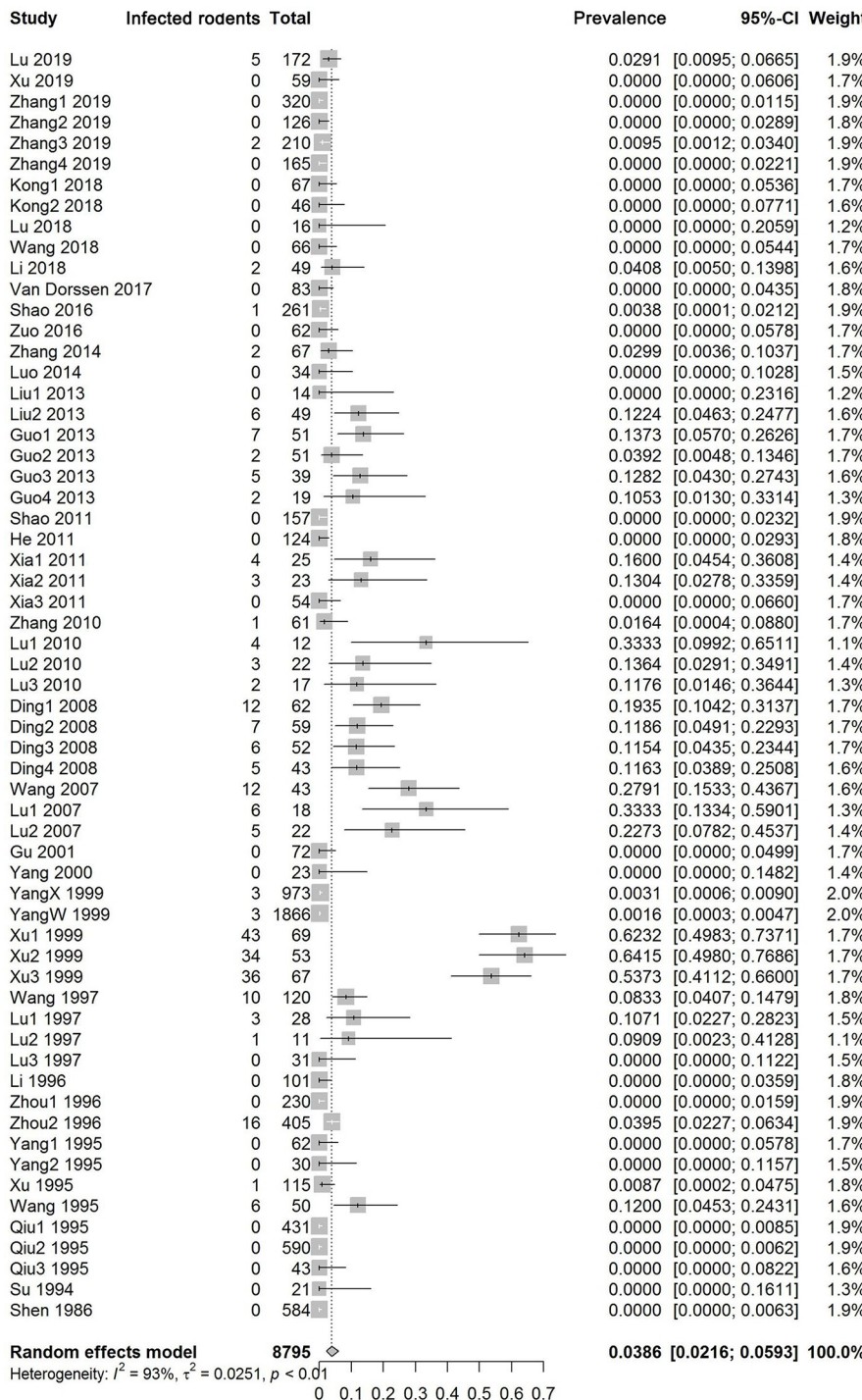

**Fig 2. Forest plot and pooled estimates of *S. japonicum* infection prevalence in rodents across China.** The diamond delimits the 95% confidence interval (95% CI) of a random effects model. The included studies were ordered by year of publication.

autumn (1.80–3.16%), and peaked in winter (22.39%). The estimate significantly (p = 0.04) increased with the density of rodents. See Table 2 & Fig 3.

**Table 2. Pooled prevalence of *S. japonicum* infections in wild rodents across China or by subgroups with meta-analysis.** Pooled prevalence was obtained via aggregation of the results of multiple studies by weighting the results of each study according to its variance.

| | No. of publications | No. of studies | No. of rodents examined | No. of rodents infected | Pooled prevalence (95% CI)% | Heterogeneity | | | | Egger's test | | | Subgroup difference | |
|---|---|---|---|---|---|---|---|---|---|---|---|---|---|---|
| | | | | | | Q-χ2 | Q/df | Q-P | I² (95% CI)% | t | t/df | P | Q-χ2 | P |
| **Overall** | 37 | 61 | 8795 | 260 | 3.86(2.16, 5.93) | 876.80 | 60 | <0.01 | 93.2(91.9, 94.2) | 5.2499 | 59 | <0.01 | | |
| **Study period** | | | | | | | | | | | | | | |
| 1980–2003 | 15 | 23 | 5975 | 156 | 3.90(1.27, 7.58) | 603.09 | 22 | <0.01 | 96.4(95.4, 97.1) | 2.7679 | 21 | 0.01 | 0.00 | 0.96 |
| 2004–2018 | 22 | 38 | 2820 | 104 | 3.73(1.81, 6.14) | 247.5 | 37 | <0.01 | 85.1(80.4, 88.6) | 5.5900 | 36 | <0.01 | | |
| **Eco-epidemiological settings** | | | | | | | | | | | | | | |
| LM | 20 | 33 | 2336 | 200 | 5.78(2.25, 10.52) | 492.42 | 32 | <0.01 | 93.5(91.8, 94.8) | 1.2308 | 31 | 0.23 | 9.63 | 0.01 |
| HM | 17 | 28 | 6459 | 60 | 1.13(0.27, 2.36) | 189.48 | 27 | <0.01 | 85.8(80.5, 89.6) | 5.3433 | 26 | <0.01 | | |
| **Province** | | | | | | | | | | | | | | |
| Jiangsu | 5 | 8 | 427 | 114 | 13.94(0.21, 40.08) | 267.66 | 7 | <0.01 | 97.4(96.2, 98.2) | -0.8807 | 6 | 0.41 | 61.80 | <0.01 |
| Anhui | 9 | 17 | 1123 | 65 | 9.85(3.60, 18.21) | 166.39 | 16 | <0.01 | 90.4(86.2, 93.3) | 5.1455 | 15 | 0.01 | | |
| Hunan | 3 | 7 | 878 | 32 | 3.83(0.54, 9.19) | 43.68 | 6 | <0.01 | 86.3(73.8, 92.8) | 1.0829 | 5 | 0.33 | | |
| Hubei | 9 | 13 | 791 | 34 | 2.60(0.31, 6.35) | 67.55 | 12 | <0.01 | 82.2(70.8, 89.2) | 1.1860 | 11 | 0.26 | | |
| Jiangxi | 2 | 2 | 188 | 5 | 1.62(0.05, 4.51) | 0.19 | 1 | 0.66 | 0.0 | /- | - | - | | |
| Sichuan | 3 | 3 | 246 | 1 | 0.22(0.00, 1.58) | 0.64 | 2 | 0.72 | 0.0(0.0, 67.7) | -2.5366 | 1 | 0.24 | | |
| Yunnan | 6 | 11 | 5142 | 9 | 0.02(0.00, 0.13) | 7.94 | 10 | 0.64 | 0.0(0.0, 49.9) | 0.5579 | 9 | 0.59 | | |
| **Season** | | | | | | | | | | | | | | |
| Winter | 4 | 8 | 383 | 124 | 22.39(3.28, 50.58) | 229.52 | 7 | <0.01 | 96.9(95.2, 97.9) | -0.3044 | 6 | 0.77 | 6.82 | 0.08 |
| Autumn | 4 | 5 | 217 | 8 | 1.80(0.00,7.40) | 12.84 | 4 | 0.01 | 68.9(20.0, 87.9) | -0.0365 | 3 | 0.97 | | |
| Summer | 4 | 7 | 2971 | 22 | 3.16(0.41, 7.55) | 56.96 | 6 | <0.01 | 89.5(80.8, 94.2) | 4.1840 | 5 | 0.01 | | |
| Spring | 3 | 3 | 168 | 2 | 0.58(0.00, 3.19) | 2.51 | 2 | 0.28 | 20.4(0.0, 91.7) | -0.2851 | 1 | 0.82 | | |
| **Density of rodents** | | | | | | | | | | | | | | |
| ≥10% | 8 | 9 | 2929 | 38 | 2.74(0.40, 6.49) | 91.14 | 8 | <0.01 | 91.2(85.6, 94.7) | 2.746 | 7 | 0.03 | 4.40 | 0.04 |
| <10% | 8 | 9 | 523 | 3 | 0.03(0.00, 0.70) | 5.68 | 8 | 0.68 | 0.0(0.0, 50.4) | 0.9114 | 7 | 0.39 | | |

Abbreviations: HM, Hilly and mountainous regions; LM, Lake and marshland regions. CI: Confidence interval; I²: Inverse variance index; Q-P: Cochran's P-value.

A total of 13 retrieved papers with 17 studies (S1 Table) provided information on rodent species. Among 19 rodent species investigated infections with *S. japonicum* were found in seven species. The *Rattus norvegicus* and *Rattus flavipectus* were the most common species investigated. The top five pooled prevalence ranged from 10.68% in *Rattus rattus* to 0.87% in

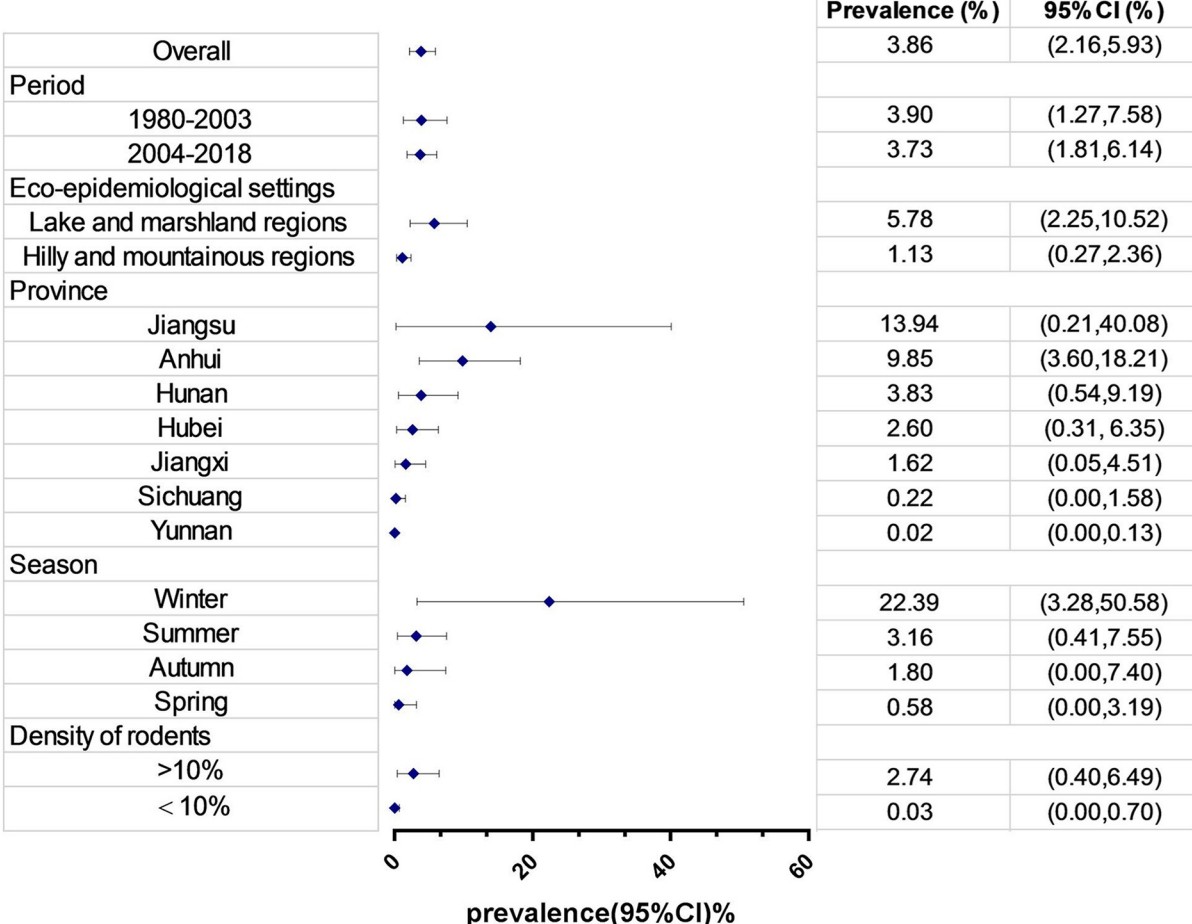

**Fig 3. Forest plot of *S. japonicum* infection prevalence in rodents pooled by subgroups.** The diamond delimits the 95% confidence interval (95% CI) of a random effects model. The forest was ordered in values of pooled schistosome prevalence.

*Apodemus sylvaticus*. No infections have been reported from three species (i.e. *Apodemus chevrier*, *Microtus fortis* and *Rattus nitidus*), whose pooled sample size each were over 100. See Table 4.

## Publication bias and sensitivity tests

Both the funnel plots (Fig 4) and the Egger linear regression test (Fig 5) indicated a potential publication bias in prevalence data (coefficient = 3.69, 95% CI: 2.29–5.10, t = 5.26, p < 0.001). The sensitivity tests showed that all single-study-omitted estimates were within the 95% CI of the respective overall infection prevalence (S1 Fig). This suggested that the pooled estimate was not substantially influenced by any single study, and hence validated the rationality and reliability of our analyses.

## Discussion

We present here the first meta-analysis of *S. japonicum* infection in wild rodents within China, incorporating data obtained from across five databases, 37 relevant articles involving 61 field studies with eligible data from 8,795 rodents. The overall mean infection prevalence of *S. japonicum* in rodents was found to be 3.86%, with no significant change between the two periods

**Table 3. Pooled prevalence of infections in rodents sub-grouped by regions and study period.** Pooled prevalence was obtained via aggregation of the results of multiple studies by weighting the results of each study according to its variance. For example, the pooled prevalence of HM in 1980–2003 was 0.00026% (written as 0.00).

| | Period | No. of studies | No. rodents examined | Pooled prevalence (%) | 95% CI (%) | Q | tau^2 | I^2(%) | Subgroup difference Q-χ2 P | |
|---|---|---|---|---|---|---|---|---|---|---|
| **Eco-epidemiology setting** | | | | | | | | | | |
| LM | 1980–2003 | 15 | 1301 | 8.85 | (1.64, 19.99) | 393.36 | 0.0858 | 96.4 | 1.56 | 0.21 |
| | 2004–2018 | 18 | 1035 | 3.56 | (1.17, 6.90) | 86.86 | 0.0179 | 80.4 | | |
| HM | 1980–2003 | 8 | 4674 | 0.00 | (0.00, 0.07) | 6.51 | 0.0000 | 0.00 | 26.03 | <0.01 |
| | 2004–2018 | 20 | 1785 | 3.93 | (1.27, 7.61) | 148.22 | 0.0200 | 87.2 | | |
| **Province** | | | | | | | | | | |
| Jiangsu | 1980–2003 | 6 | 304 | 21.43 | (1.00, 55.35) | 188.84 | 0.1842 | 97.4 | 4.30 | 0.04 |
| | 2004–2018 | 2 | 123 | 0.51 | (0.00, 3.13) | 1.03 | <0.0001 | 2.2 | | |
| Hubei | 1980–2003 | 2 | 122 | 0.00 | (0.00, 1.03) | 0.24 | 0.0000 | 0.00 | 4.98 | 0.03 |
| | 2004–2018 | 11 | 669 | 3.41 | (0.53, 8.01) | 60.11 | 0.0207 | 83.40 | | |
| Hunan | 1980–2003 | 2 | 635 | 1.26 | (0.00, 7.81) | 17.00 | 0.0136 | 94.10 | 1.83 | 0.18 |
| | 2004–2018 | 5 | 243 | 6.17 | (0.58, 16.65) | 20.33 | 0.0218 | 80.3 | | |
| Anhui | 1980–2003 | 6 | 824 | 4.24 | (0.00, 14.03) | 58.35 | 0.0341 | 91.40 | 2.65 | 0.10 |
| | 2004–2018 | 11 | 299 | 13.82 | (6.21, 23.42) | 40.00 | 0.0278 | 75.00 | | |
| Yunnan | 1980–2003 | 5 | 3903 | 0.01 | (0.00, 0.11) | 3.08 | 0.0000 | 0.00 | 0.73 | 0.39 |
| | 2004–2018 | 6 | 1239 | 0.10 | (0.00, 0.46) | 4.12 | 0.0000 | 0.00 | | |
| Jiangxi | 1980–2003 | 0 | | - | - | - | - | - | | |
| | 2004–2018 | 2 | 188 | 1.62 | (0.05, 4.51) | 0.91 | 0.0000 | 0.00 | | |
| Sichuan | 1980–2003 | 2 | 187 | 0.38 | (0.00, 2.13) | 0.51 | 0.0000 | 0.00 | 0.13 | 0.72 |
| | 2004–2018 | 1 | 59 | 0.00 | (0.00, 2.89) | 0.00 | - | - | | |

Abbreviations: HM, Hilly and mountainous regions; LM, Lake and marshland regions.

of 1980 to 2003 and 2004 to 2018, i.e. before and after the integral control strategy was implemented [35], at 3.90% and 3.73% respectively. Such an observed lack of apparent reduction in overall pooled *S. japonicum* infection prevalence amongst rodent species between the two time

**Table 4. Pooled prevalence of *S. japonicum* infections in rodents by species with meta-analysis.** Pooled prevalence was obtained via aggregation of the results of multiple studies by weighting the results of each study according to its variance.

| Species* | No. of studies | No. of rodents examined | No. of rodents infected | Pooled prevalence (%) | 95% CI (%) | Q | tau^2 | I^2(%) |
|---|---|---|---|---|---|---|---|---|
| Overall | 17 | 3970 | 185 | 0.80 | (0.00, 3.21) | 628.85 | 0.0401 | 90.8 |
| *Rattus rattus* | 5 | 78 | 12 | 10.68 | (2.46, 22.47) | 4.83 | 0.0038 | 17.4 |
| *Rattus losea* | 1 | 41 | 4 | 9.76 | (2.20, 21.07) | 0 | - | - |
| *Rattus norvegicus* | 14 | 1226 | 150 | 5.95 | (0, 17.91) | 410.35 | 0.0968 | 96.8 |
| *Rattus flavipectus* | 10 | 1380 | 13 | 1.58 | (0, 7.89) | 43.73 | 0.0217 | 79.4 |
| *Apodemus sylvaticus* | 1 | 115 | 1 | 0.87 | (0, 3.70) | 0 | - | - |
| *Rattus sladeni* | 3 | 302 | 2 | 0.43 | (0, 1.77) | 1.45 | 0 | 0.0 |
| *Apodemus agrarius* | 5 | 196 | 3 | 0.39 | (0, 3.37) | 6.76 | 0.0045 | 40.8 |
| *Microtus fortis* | 7 | 198 | 0 | 0 | (0, 0.10) | 1.18 | 0 | 0.0 |
| *Apodemus chevrier* | 3 | 143 | 0 | 0 | (0, 1.34) | 0.07 | 0 | 0.0 |
| *Rattus nitidus* | 1 | 237 | 0 | 0 | (0, 0.72) | 0 | - | - |

* Rodent species with pooled sample size of less than 20 are not listed, including *Eothenomys miletus, Microtus montanus, Micromys minutus, Mus musculus, Mus pahari, Mus caroli. Rattus yunnanensis, Rattus koraten* and *Suncus murinus*.

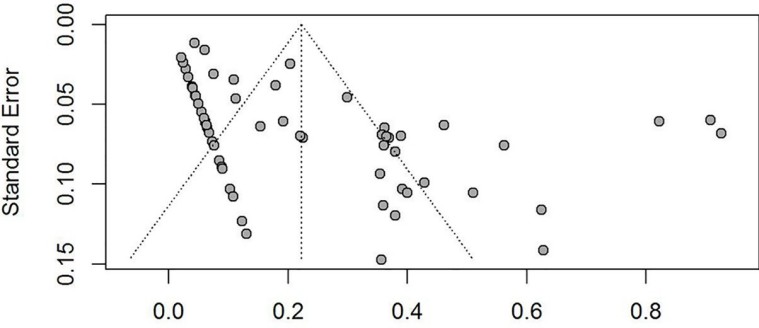

**Fig 4. Funnel plots of the Freeman-Tukey double arcsine transformed prevalence of *S. japonicum* infection in rodents.** The vertical lines and diagonal dashed lines represent the overall estimated effect size and its 95% confidence limits, respectively. Each dot represents a different study.

periods contrasts starkly with the significant reduction in infections reported in humans and livestock over these time periods [5, 7]. The estimates did, however, significantly vary in relation to region or season, as well with rodent density or species. Estimates from lake and marshland regions showed a non-significant decline following the increased control pressures imposed post 2004, whereas infection prevalence levels in rodents significantly increased within the hilly and mountainous regions. This is consistent with the most recent

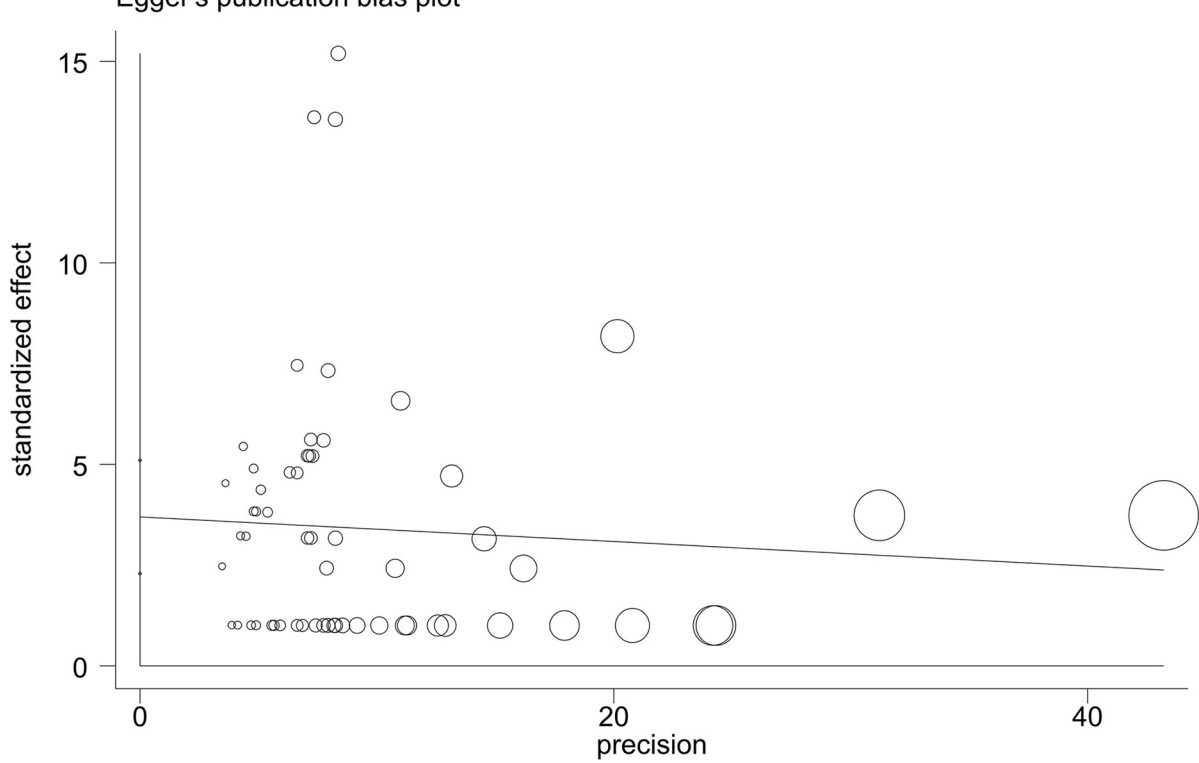

**Fig 5. Egger's publication bias plot of the included studies of the effect of *S. japonicum* infection prevalence in rodents.** The size of circles indicates the sample size of an individual study.

epidemiological and chronobiological data, combined with mathematical models, all identifying rodents as current key hosts responsible for maintaining *S. japonicum* transmission within hilly/mountainous regions, whilst bovines as key hosts only within lake and marshland regions [17, 22].

Some possible explanations for this contrasting trajectory by habitat type may be proposed. As specified above, in 2004, the central government of China classified schistosomiasis as one of the highest priorities in infectious diseases control [26] and then developed the medium- and long-term control plan [27]. As most infections in humans were observed at the time to be occurring within lake and marshland regions, and bovines had been shown to contribute a large part to the local transmission in such areas [14, 15], an integrated control strategy which emphasized mechanization of agriculture, fencing of bovines, access to clean water and adequate sanitation, health education, focal snail control, along with chemotherapy of both humans and bovines was successfully implemented in four pilot areas during 2004 to 2008 [35]. Since then the new strategy has been widely applied, and it has proved highly effective particularly in the lake and marshland areas of China, as reflected by the rapid reduction in number of infection in humans, bovines and snail intermediate hosts reported [73]. However, in certain mountainous/hilly areas where, conversely, wild animal key host reservoirs for *S. japonicum* exist, such strategies were less logistically feasible, consistent with the lowered relative effectiveness [16, 17, 74].

Here we observed that the pooled prevalence of *S. japonicum* identified in rodents also varied greatly among provinces in the middle and lower reaches of the Yangtze River, with a significant reduction in Jiangsu after 2004 but an increase in other three provinces (i.e. Hubei, Hunan and Anhui). Even in Hubei in 2016, although no infections in humans or bovines, nor indeed amongst snail intermediate hosts, were found across the whole region, infected rodents were still identified [43]. The uneven development in economy and the existence of complicated environments might be the main explanations. Jiangsu province was one of the most heavily endemic regions of schistosomiasis with the majority belonging to the lake and marshland regions. Since 1980s the social and economic reformation has resulted in strong economic growth for the province. This may have accelerated its work in schistosomiasis control as the implication of the new integrated strategy, for example mechanization of farming is resource intensive. Compared to Jiangsu, all other endemic provinces were less developed and the implication of the strategy might be jeopardized or even hampered. It must be acknowledged, however, that the numbers of studies performed varied among provinces and/or periods, with, for example, during 1980 to 2003 only eight and two studies were performed in the hilly and mountainous regions and in Hubei province respectively. Where fewer studies were performed, one could reasonably propose that this may largely reflect the local professional awareness (or lack of) regarding schistosomiasis transmission and control in these regions and habitats. Indeed, besides the long-neglected awareness of the rodents' role in transmission, the endemic status in the hilly and mountainous regions was generally considered to be less serious than in the lake and marshland regions, particularly since, for example in 2003, the ratio of snail-infested area in the former to the latter was 1 to 21.6 [5]. However, we do also acknowledge that small study numbers could possibly result in a potential bias in our inferred low estimates here and that further research here is certainly warranted.

The prevalence estimate amongst rodents overall also changed over seasons of a year, increasing from spring (0.58%), throughout summer and autumn (1.80–3.16%), and to winter (22.39%). The upward trend mirrors the scenario that most infections occur during the seasons when temperatures are more suitable for schistosome cercariae to be released into water [75, 76]. Cumulative infections of schistosomes in rodents over time, if they continually frequent in infested areas, could be reasonably assumed.

It has been reported that different species of rodents show a wide range in susceptibility to *S. japonicum*, from fully permissive to non-permissive for schistosome infection [77, 78], although *Microtus fortis* are the only mammals in which it has been experimentally confirmed to be non-permissive to schistosome infection [79, 80]. In our meta-analyses the pooled prevalence varied among rodent species, with the highest estimate at 10.68% in *Rattus rattus* and zero infections in *M. fortis*. *Rattus norvegicus* and *Rattus flavipectus* were most often investigated. They showed prevalence of 5.95% and 1.58% respectively, both being higher than the proposed threshold of 1% for schistosome interruption [81], although *Rattus norvegicus* was once believed to be less susceptible [77].

In addition to infection prevalence within rodents, it is also critical to stress the, often very high, infection intensities reported within such rodents and that there is convincing evidence that those eggs shed are viable and hence contributing to ongoing transmission. For example, Mao reported that the mean infection intensity was 62.06 miracidia per gram of rodent (*Rattus norvegicus*) faeces [82]. Moreover, Lu and colleagues studying infected rodents from hilly villages during 2006–2007 reported mean infection intensities (hatched miracidia plus eggs per gram of faeces) in these regions of up to 231 [54]. Furthermore, evidence of the viability of these parasites shed by rodents and their involvement within ongoing transmission across a broader multi-host spectrum has been provided by molecular studies typing these hatched miracidia and revealing often large proportions of schistosome genotypes shared between rodents and those obtained from miracdia hatched from humans, as well as other potential key definitive hosts such as dogs [74]. Similar findings have also recently reported for *S. mansoni* in West Africa, in which viable *S. mansoni* miracidia were collected from rodent stool and matched schistosome genotypes were found between humans and rodents indicative of shared ongoing transmission within certain regions or habitats [23].

Small wild rodents have often been neglected in relation to schistosomiasis control globally. The estimated high prevalence and intensities levels, even in previously believed to be 'less susceptible' species, their increases in parallel with rodent densities, and the shared genotypes between host species, all suggest that infected rodents could be of significant importance in maintaining transmission of the parasite in some areas within China where rodent density is considerably high. Indeed, the potential role of rodents in the continued maintenance for other human schistosome species is also gathering credence globally–from, for example, the insular Guadeloupean focus rodents known to maintain local transmission for *S. mansoni* [83] to their role as reservoir hosts and/or biotic hubs for ongoing transmission of both *S. mansoni* and also potentially *S. haematobium* group hybrids in both Africa [23, 84] and even Europe [85].

We do, of course, fully acknowledge the potential inherent limitations within our study. First, there is the potential publication bias in our research. We identified very few publications from either Jiangxi or Sichuan, although these were two of the most serious endemic provinces [5]. However, as the assessment framework of schistosome transmission in an area only currently reports instances amongst humans, domestic livestock and/or snail intermediate hosts [86], with investigations of *S. japonicum* infections in rodents not being obligatory, this inevitably leads to no or few studies in some areas, and likewise a potential publication bias pre 2004. This could be the case in 1980–2003 in the hilly and mountainous regions and in Hubei province, each with low pooled prevalence estimated. Another important issue is that few papers reported information on infection intensity in rodents, and hence it was not possible to calculate relative indices between species within our meta-analyses [87]. Finally, a problem inherent in many meta-analysis studies in general, sample size varied greatly from study to study. Nevertheless, in terms of our data here, this appeared primarily due to the existence of rodent density differences between areas, seasons and hence studies–and as we performed

subgroup analyses based on the density of rodents and found a positive correlation between both, we remain confident in the power of our analyses on the estimates of infections in wild rodents across China.

## Conclusions

This study systematically analyzed the available literature on *S. japonicum* infections in rodents over the last forty years after the introduction of praziquantel for schistosomiasis treatment in humans and livestock. Although infections in both humans and bovines have been on a downward trend [7, 8], in line with the new targets for elimination of schistosomiasis within China [9], in stark contrast, our meta-analyses indicate that no such concurrent decline has been observed amongst rodent infections. Moreover, whilst estimates varied by area, season and rodent density, there was a significant upward trend towards an increased prevalence over time of *S. japonicum* amongst rodents in hilly regions. This is compatible with recent epidemiological and mathematical models which indicated that rodents may be sufficient to maintain ongoing transmission within certain hilly/mountainous regions at least [17]. Furthermore, in terms of monitoring and evaluation of ongoing disease control and elimination programmes, particularly where regions and/or countries may require official WHO verification of interruption of transmission, such studies to date have already identified areas where no infections in humans and livestock were reported, but infections in rodents were still identified (e.g. [43]). Thus, we stress here the imperative need for future systematic research in this area, and if subsequently confirmed necessary, that formalized monitoring (ideally involving both parasitological and molecular tools) amongst rodent wildlife populations be implemented at the final stages of any 'elimination' evaluation. Rodents, among mammals, are the most abundant and include the greatest number of zoonotic hosts (approximately 10.7% of species, carrying 85 unique zoonotic diseases) [88]. Accordingly, rodents have been projected to become the dominant wildlife in human-driven environments and the main reservoir of zoonotic diseases in tropical zones [89]. The extent to which rodents contribute to the zoonotic transmission of schistosomiasis remains an essential question to be further developed by ecological and epidemiological approaches, genetics and genomics, together with mathematical modelling combined. As current efforts aim towards interruption of schistosomiasis transmission, the potential implications of alternative hosts such as rodents in the disease dynamics should not be ignored. Any rebounds of schistosomiasis may threaten to undermine future public health, and indeed One Health, interventions across, regional, national and international scales.

## Supporting information

**S1 Checklist. PRISMA checklist.**
(DOCX)

**S1 Table. *S. japonicum* infections in different species of rodents from 13 articles.**
(DOCX)

**S1 Fig. The sensitivity analysis of *S. japonicum* infection prevalence in rodents for all studies.**
(TIF)

## Author Contributions

**Conceptualization:** Hui-Ying Zou, Da-Bing Lu.

**Data curation:** Hui-Ying Zou, Qiu-Fu Yu.

**Formal analysis:** Hui-Ying Zou, Qiu-Fu Yu.

**Funding acquisition:** Joanne P. Webster, Da-Bing Lu.

**Methodology:** Hui-Ying Zou, Qiu-Fu Yu, Chen Qiu.

**Project administration:** Da-Bing Lu.

**Resources:** Hui-Ying Zou, Chen Qiu.

**Software:** Hui-Ying Zou, Chen Qiu.

**Supervision:** Joanne P. Webster, Da-Bing Lu.

**Validation:** Qiu-Fu Yu, Chen Qiu, Joanne P. Webster, Da-Bing Lu.

**Writing – original draft:** Hui-Ying Zou.

**Writing – review & editing:** Joanne P. Webster, Da-Bing Lu.

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
