## [Decision Letter · Decision Letter 0]

5 Feb 2020

Dear Dr. Lu,

Thank you very much for submitting your manuscript "A meta-analysis of prevalence of Schistosoma japonicum infections in wild rodents across China after introduction of praziquentel for a large-scale treatment of schistosomiasis" for consideration at PLOS Neglected Tropical Diseases. As with all papers reviewed by the journal, your manuscript was reviewed by members of the editorial board and by several independent reviewers. In light of the reviews (below this email), we would like to invite the resubmission of a significantly-revised version that takes into account the reviewers' comments.

We cannot make any decision about publication until we have seen the revised manuscript and your response to the reviewers' comments. Your revised manuscript is also likely to be sent to reviewers for further evaluation.

Sincerely,

Justin V. Remais, Ph.D.

Deputy Editor

Reviewer's Responses to Questions

**Key Review Criteria Required for Acceptance?**

**Methods**

-Are the objectives of the study clearly articulated with a clear testable hypothesis stated?

-Is the study design appropriate to address the stated objectives?

-Is the population clearly described and appropriate for the hypothesis being tested?

-Is the sample size sufficient to ensure adequate power to address the hypothesis being tested?

-Were correct statistical analysis used to support conclusions?

-Are there concerns about ethical or regulatory requirements being met?

Reviewer #1: (No Response)

Reviewer #2: This study systematically analyzed S. japonicum infections in wild rodents from the

published literature over the last forty years after the introduction of praziquantel for

schistosomiasis treatment in humans and bovines in 1980s. The analysis seems appropriate.

Reviewer #3: (No Response)

**Results**

-Does the analysis presented match the analysis plan?

-Are the results clearly and completely presented?

-Are the figures (Tables, Images) of sufficient quality for clarity?

Reviewer #1: (No Response)

Reviewer #2: Although numbers of schistosomiasis cases in humans and bovines have been greatly reduced in China, change of infection prevalence in rodents differed among provinces or eco-epidemiology settings.

In Table 1, the author and year of publication do not match for many of the studies highlighted.

Reviewer #3: (No Response)

**Conclusions**

-Are the conclusions supported by the data presented?

-Are the limitations of analysis clearly described?

-Do the authors discuss how these data can be helpful to advance our understanding of the topic under study?

-Is public health relevance addressed?

Reviewer #1: (No Response)

Reviewer #2: While China is determining to achieve the target of complete interruption of zoonotic

schistosomiasis in the country as a whole by 2030, the authors conclude that enrolling an investigation of S. japonicum infections in small animal rodents into the ongoing monitoring and

evaluation program is essential to determine whether local transmission is ongoing.

There are a number of issues that the authors need to consider carefully as their conclusions as presented are somewhat misleading and could impact negatively (although unlikely as the programme is well advanced) on the national control programme for schistosomiasis in China.

It is not clear how prevalence was obtained - dissection of animals or detection of eggs in faeces.

There are a number of rodent species in China and it would be important to include these if known from the studies considered as different species of rodents range from fully permissive to non-permissive for schistosome infections. Similarly it is unclear if any of the quoted studies include intensity information which is unfortunate as the Index of Potential Contamination (Vercruysse et al. Trends Parasitol. 17, 256-61, 2001) (important for defining schistosome transmission parameters) cannot be calculated. A recent article by Van Dorssen et al. Parasitology 2017, 144: 1633-42) showed that the Contamination index for rodents (two species) in the Dongting Lake area of Hunan province was extremely low. 

Furthermore, the authors do not provide any information for any of the studies on the maturity of worms in infected rodents or the viability of eggs other than quoting an older article by Mao (1990) stating "The egg hatching rate from an infected rodent's feces was 14.5 times that of human feces". This statement needs to be revisited as it does not seem to be realistic and needs additional experimental evidence in support, particularly as several different rodent species, with different susceptibilities to infection, will be present in the endemic areas.

Reviewer #3: (No Response)

**Editorial and Data Presentation Modifications?**

Reviewer #1: (No Response)

Reviewer #2: English style and grammar needs considerable attention. Praziquantel is misspelt on numerous occasions. In Table 1, the author and year of publication do not match for many of the studies highlighted. References do not conform to the correct journal style.

Reviewer #3: (No Response)

**Summary and General Comments**

Reviewer #1: Needs to be checked for English. Perhaps the fourth author, Joanne, could do this.

Line 65: seemed is past tense, presumably it is imperative 

Line 70: transmission interruption (and would a lay person know what this means?)

Line 209: Differences in how infection status was evaluated?

Line 230: Space before (

There have been several papers in the past throwing doubt as to the permissibility of rodents as schistosome hosts – i.e. that they do not have viable infections. Any comment on this? Were viability measures included in the studies examined? What methods were used to determine infection? Eggs in stool? Conceivably rodents could be consuming the eggs in the faeces of humans or other animals and passing them through the stool? 

I am aware of at least one paper on rodents in China that has not been included; Van Dorssen et al 2017, Para. 144 (12):163-1642. 

Do you think your search terms were too restrictive and a more open ended search would have yielded additional papers?

Reviewer #2: This authors need to temper their conclusions in light of the review comments made above.

The title of the article does not convey the true content of the paper.

Reviewer #3: This study tried to estimate the infection prevalence of S. japonicum in rodents in China by meta-analysis, and it is interesting. However, people used to pay attention to infections in humans and livestock, and in snail intermediate hosts, and to ignore S. japonicum infections in rodents or other wild animals. As a result, there might be not enough studies to reflect the epidemic situation of rodents in China. As the author said, few publications on studies were performed in Jiangxi and Sichuan, two of the most serious endemic provinces. 

Hence, caution is needed in making conclusions. For example, in Table 3, the prevalence of rodents in hilly and mountainous regions was 0 during 1980-2003. Is it really the case? And is it real that ‘the estimate significantly increased in the hilly and mountainous regions’. There is also no data to support that “The estimate significantly decreased over time in the marshland and lakes”, for the 95% CI of prevalence during 1980-2003 is (0.0164, 0.1999), and that during 2004-2018 is (0.0145, 0.0834) (Table 3).

PLOS authors have the option to publish the peer review history of their article (what does this mean?). If published, this will include your full peer review and any attached files.

Reviewer #1: No

Reviewer #2: No

Reviewer #3: No
---

## [Decision Letter · Decision Letter 1]

19 May 2020

Dear Dr. Lu,

Thank you very much for submitting your manuscript "Meta-analyses of Schistosoma japonicum infections in wild rodents across China over time indicate persistence: an ultimate challenge to the 2030 elimination targets" for consideration at PLOS Neglected Tropical Diseases. As with all papers reviewed by the journal, your manuscript was reviewed by members of the editorial board and by several independent reviewers. In light of the reviews (below this email), we would like to invite the resubmission of a significantly-revised version that takes into account the reviewers' comments.

We cannot make any decision about publication until we have seen the revised manuscript and your response to the reviewers' comments. Your revised manuscript is also likely to be sent to reviewers for further evaluation.

Sincerely,

Darren J. Gray

Associate Editor

Justin Remais

Deputy Editor

Reviewer's Responses to Questions

**Key Review Criteria Required for Acceptance?**

**Methods**

-Are the objectives of the study clearly articulated with a clear testable hypothesis stated?

-Is the study design appropriate to address the stated objectives?

-Is the population clearly described and appropriate for the hypothesis being tested?

-Is the sample size sufficient to ensure adequate power to address the hypothesis being tested?

-Were correct statistical analysis used to support conclusions?

-Are there concerns about ethical or regulatory requirements being met?

Reviewer #1: (No Response)

Reviewer #2: (No Response)

**Results**

-Does the analysis presented match the analysis plan?

-Are the results clearly and completely presented?

-Are the figures (Tables, Images) of sufficient quality for clarity?

Reviewer #1: (No Response)

Reviewer #2: (No Response)

**Conclusions**

-Are the conclusions supported by the data presented?

-Are the limitations of analysis clearly described?

-Do the authors discuss how these data can be helpful to advance our understanding of the topic under study?

-Is public health relevance addressed?

Reviewer #1: (No Response)

Reviewer #2: (No Response)

**Editorial and Data Presentation Modifications?**

Reviewer #1: (No Response)

Reviewer #2: (No Response)

**Summary and General Comments**

Reviewer #1: English is much improved, some small issues remain. See specific comments.

Abstract

Line 23: This first sentence can be worded much better. 

Line 45: ‘change was observed between before and after 2004’ – what does this mean? Between 2004 and what? No significant changes were observed, ever?

Line 45: estimated

Line 58: Are there population estimates available for rodents, or notes on plague years? Obviously very few papers on rodents and schisto, so linking to total populations in endemic areas might be useful.

Line 79: delete often and levels 

Line 80: has instead of have, and in instead of amongst

Line 81: Delete ‘of the few performed to date’

Line 84: that instead of the

Line 94: Ah, so only papers where dissection occurred? Any papers identified but not used that did not use dissection, but other methods?

Line 96: ‘change was observed between before and after 2004’ – what does this mean? Between 2004 and what? No significant changes were observed, ever?

Line 112: After Schistosoma in full for mansoni, can abbreviate to S. haematobium, and S. japonicum

Line 114: This sentence is awkward, needs to be re-written

Line 115: A the beginning of the national

Line 122: Delete for example

Line 150: provide

Line 158: Whilst

Line 177: papers

Line 194: investigations

Line 258: add the p value in brackets after significantly

Line 269: ordered by year of publication

Line 312-318 is a loooong sentence. I would put the second part of the sentence first, then follow up with, this is consistent blah blah blah, first part of sentence.

“Estimates from lake and marshland regions showed a non-significant decline following the increased control pressures imposed post 2004, whereas infection prevalence levels in rodents

significantly increased within the hilly and mountainous regions. This is consistent with mathematical models identifying rodents as current key hosts responsible for maintaining S. japonicum transmission within hilly/mountainous regions, with bovines as key hosts only within lake and marshland regions.”

Line 338: Definitely need to make the point of discrepancy’s in studies performed. Looking at table 3, Jiangsu 1980-2003 6 papers, 2004-2018 2. Not a lot of available data to make too many sweeping statements on prevalence increases and decreases without referencing that there is little info available. You then have a different bias in Hubei with 11 studies post-204, and only 2 pre 2004. Obviously it is what it is, but it needs to be acknowledged here.

Line 318: identified very few

Table 3: May be useful to put the number of rodents in. Particularly when looking at the prev. I noticed looking at tbale 2 that while there were more studies in 2004-2018 than pre-2004, there were less rodents examined. So that is important info to have in table 3 for this comparison as well.

Reviewer #2: I appreciate the amount of work and effort that the authors have undertaken but this reviewer is still not convinced about the major conclusion that wild rodents contribute substantially to transmission of S. japonicum and that they will be a major barrier preventing 2030 elimination targets in China. The message presented is too strong and this reviewer does not subscribe to their conclusions for the earlier reasons I presented. Table 1 provides useful information but again there is no detail on the viability of eggs in the infected animals. It is also unclear what the % density of rodents means.

As China nears elimination of schistosomiasis and the prevalence in humans and bovines is extremely low approaching zero (see Reference 8 they quote), if rodents play a key role in transmission, why arent these prevalence values higher? 

This reviewer suggests a major rewrite of the article just emphasising the valuable retrospective information presented in Table 1 (indicating the flaws in many of these papers as parameters such as intensity levels, viability/hatchability of eggs were not considered) and de-emphasising the impact on China's path to schistosomiasis elimination.

PLOS authors have the option to publish the peer review history of their article (what does this mean?). If published, this will include your full peer review and any attached files.

Reviewer #1: No

Reviewer #2: No
---

## [Editor Report · Decision Letter 2]

19 Jun 2020

Dear Dr. Lu,

Thank you very much for submitting your manuscript "Meta-analyses of Schistosoma japonicum infections in wild rodents across China over time indicates a potential challenge to the 2030 elimination targets" for consideration at PLOS Neglected Tropical Diseases. As with all papers reviewed by the journal, your manuscript was reviewed by members of the editorial board and by several independent reviewers. In light of the reviews (below this email), we would like to invite the resubmission of a significantly-revised version that takes into account the reviewers' comments. 

Thanks for toning down the manuscript and for addressing the reviewer comments. I have some additional comments: 

1. I think it is import that there is explicit discussion on the ability of rodents to produce viable eggs. You rebutted the reviewer but this needs to go in the manuscript. The discussion should be along lines of do rodents produce schistosome eggs that hatch into infectious miracidia - this should be supported with evidence 

2. Greater detail on the pooling of prevalence across the studies needs to be provided in the methods. As it stands in tables 2 & 3 with the provision of total number of rodents and total infected rodents the prevalence does not add up - this is misleading and so the pooled prevalence requires greater explanation

3. Greater discussion is required around the increase of prevalence between the 2 time periods in the hill and mountainous region. The prevalence was zero in the 1980 - 2003 period and this is what is driving the significant increase seen in 2004 - 2018. Is this zero estimate correct? significant transmission was ongoing in the 1980-2003 period and I find it strange that the prevalence is zero, noting it is 8% in the lake and marshlands during the same period. only 8 studies were undertaken could it be that this was not enough to capture a true estimate of prevalence. This requires considerable discussion as this result underpins your argument. the same applies to Hubei.

We cannot make any decision about publication until we have seen the revised manuscript and your response to the reviewers' comments. Your revised manuscript is also likely to be sent to reviewers for further evaluation.

Sincerely,

Darren J. Gray

Associate Editor

Justin Remais

Deputy Editor

Thanks for toning down the manuscript and for addressing the reviewer comments. I have some additional comments: 

1. I think it is import that there is explicit discussion on the ability of rodents to produce viable eggs. you rebutted the reviewer but this needs to go in the manuscript. The discussion should be along lines of do rodents produce schistosome eggs that hatch into infectious miracidia - this should be supported with evidence 

2. Greater detail on the pooling of prevalence across the studies needs to be provided in the methods. As it stands in tables 2 & 3 with the provision of total number of rodents and total infected rodents the prevalence does not add up - this is misleading and so the pooled prevalence requires greater explanation

3. Greater discussion is required around the increase of prevalence between the 2 time periods in the hill and mountainous region. The prevalence was zero in the 1980 - 2003 period and this is what is driving the significant increase seen in 2004 - 2018. Is this zero estimate correct? significant transmission was ongoing in the 1980-2003 period and I find it strange that the prevalence is zero, noting it is 8% in the lake and marshlands during the same period. only 8 studies were undertaken could it be that this was not enough to capture a true estimate of prevalence. This requires considerable discussion as this result underpins your argument. the same applies to Hubei.
---

## [Editor Report · Decision Letter 3]

27 Jul 2020

Dear Dr. Lu,

We are pleased to inform you that your manuscript 'Meta-analyses of Schistosoma japonicum infections in wild rodents across China over time indicates a potential challenge to the 2030 elimination targets' has been provisionally accepted for publication in PLOS Neglected Tropical Diseases.

Best regards,

Darren J. Gray

Associate Editor

Justin Remais

Deputy Editor

---

## [Editor Report · Acceptance letter]

26 Aug 2020

Dear Dr. Lu,

We are delighted to inform you that your manuscript, "Meta-analyses of Schistosoma japonicum infections in wild rodents across China over time indicates a potential challenge to the 2030 elimination targets," has been formally accepted for publication in PLOS Neglected Tropical Diseases.

Best regards,

Shaden Kamhawi

co-Editor-in-Chief

Paul Brindley

co-Editor-in-Chief
